# Chemokinergic and Dopaminergic Signalling Collaborates through the Heteromer Formed by CCR9 and Dopamine Receptor D5 Increasing the Migratory Speed of Effector CD4^+^ T-Cells to Infiltrate the Colonic Mucosa

**DOI:** 10.3390/ijms251810022

**Published:** 2024-09-18

**Authors:** Javier Campos, Francisco Osorio-Barrios, Felipe Villanelo, Sebastian E. Gutierrez-Maldonado, Pablo Vargas, Tomás Pérez-Acle, Rodrigo Pacheco

**Affiliations:** 1Centro Científico y Tecnológico de Excelencia Ciencia & Vida, Fundación Ciencia & Vida, Huechuraba 8580704, Santiago, Chile; jvrcamposa@gmail.com (J.C.); felipe@dlab.cl (F.V.); sebastian.gutierrez@uss.cl (S.E.G.-M.); tperezacle@cienciavida.org (T.P.-A.); 2Gut Microbiology, Institute for Infectious Diseases, University of Bern, Friedbühlstrasse 25, 3001 Bern, Switzerland; 3Escuela de Ingeniería, Facultad de Ingeniería Arquitectura y Diseño, Universidad San Sebastián, Recoleta 8420524, Santiago, Chile; 4Institut Curie, PSL Research University, CNRS, UMR144, F-75005 Paris, France; pablo.vargas@inserm.fr; 5Université Paris Cité, CNRS, INSERM, Inserm, INEM, F-75015 Paris, France; 6Facultad de Medicina y Ciencia, Universidad San Sebastián, Providencia 7510157, Santiago, Chile

**Keywords:** dopamine, chemokines, T-cell migration, G protein-coupled receptors, heteromers, inflammatory bowel diseases

## Abstract

Inflammatory bowel diseases (IBDs) involve chronic inflammation of the gastrointestinal tract, where effector CD4^+^ T-cells play a central role. Thereby, the recruitment of T-cells into the colonic mucosa represents a key process in IBD. We recently found that CCR9 and DRD5 might form a heteromeric complex on the T-cell surface. The increase in CCL25 production and the reduction in dopamine levels associated with colonic inflammation represent a dual signal stimulating the CCR9:DRD5 heteromer, which promotes the recruitment of CD4^+^ T-cells into the colonic lamina propria. Here, we aimed to analyse the molecular requirements involved in the heteromer assembly as well as to determine the underlying cellular mechanisms involved in the colonic tropism given by the stimulation of the CCR9:DRD5 complex. The results show that dual stimulation of the CCR9:DRD5 heteromer potentiates the phosphorylation of the myosin light chain 2 (MLC2) and the migration speed in confined microchannels. Accordingly, disrupting the CCR9:DRD5 assembly induced a sharp reduction in the pMLC2 in vitro, decreased the migratory speed in confined microchannels, and dampened the recruitment of CD4^+^ T-cells into the inflamed colonic mucosa. Furthermore, in silico analysis confirmed that the interface of interaction of CCR9:DRD5 is formed by the transmembrane segments 5 and 6 from each protomer. Our findings demonstrated that the CCR9:DRD5 heteromeric complex plays a fundamental role in the migration of CD4^+^ T-cells into the colonic mucosa upon inflammation. Thereby, the present study encourages the design of strategies for disassembling the formation of the CCR9:DRD5 as a therapeutic opportunity to treat IBD.

## 1. Introduction

Inflammatory bowel diseases (IBDs) form a group of chronic remittent inflammatory disorders of the gastrointestinal tract, among which Crohn’s disease (CD) and ulcerative colitis (UC) are the most common. The overall IBD prevalence is approximately five million people worldwide, which has been increasing during the last decade [1]. Evidence from animal models and IBD patients has indicated that effector CD4^+^ T-cells, including the subsets Th1 and Th17, play a central role in this chronic inflammation [2]. Thereby, the recruitment of T-cells into the colonic mucosa represents a key process in IBD.

The C-C chemokine receptor 9 (CCR9) constitutes a key homing molecule that leads the recruitment of T-cells into the small intestine mucosa under homeostatic conditions [3]. Indeed, CCR9 plays an important role in oral tolerance by recruiting regulatory T-cells (Treg) into the small intestine lamina propria [4]. Nevertheless, the production of the endogenous CCR9 agonist, CCL25, is highly enhanced in the colonic mucosa upon inflammation, inducing the infiltration of CCR9^+^ inflammatory T-cells in the colonic lamina propria [5]. Thus, whereas CCR9 drives the trafficking of T-cells into the small intestine in the absence of inflammation, this molecule favours the infiltration of inflammatory lymphocytes into the colonic mucosa upon inflammation.

Importantly, the colonic mucosa represents one of the major sources of dopamine, a key regulator of inflammation [6,7]. Under homeostatic conditions, dopamine is found at high concentrations in the colonic mucosa in humans [8] and mice [9,10], which might stimulate both high-affinity (DRD3 and DRD5) and low-affinity (DRD1 and DRD2) dopamine receptors. Although the signalling of DRD1 and DRD2 is dominant over DRD3 and DRD5 at high dopamine levels, the molecular mechanisms underlying this process remain unexplored. The stimulation of both DRD1 and DRD2 has been associated with anti-inflammatory down-stream effects [11,12,13,14,15], thereby favouring immune tolerance. However, colonic dopamine levels are strongly decreased upon intestinal inflammation in IBD patients [8] as well as in animal models [9]. This change in the concentration of colonic dopamine involves a switch in the dopamine receptors stimulated, favouring the selective stimulation of high-affinity dopamine receptors, including DRD3 [16] and DRD5 [7]. Indeed, the DRD3 signalling on T-cells promotes the Th1 and Th17 function [17], while dampening the suppressive activity of Treg [18], thus favouring gut inflammation. Interestingly, the stimulation of DRD3 in Treg also down-regulates the CCR9 expression, thus impairing the recruitment of Treg to the colonic mucosa and exacerbating the inflammatory process [18]. Moreover, DRD5 stimulation potentiates the immunogenic profile of dendritic cells [19,20,21] and favours Th17 responses [21,22], thereby contributing to inflammation too.

We previously found that CCR9 and DRD5 might form an heteromeric complex on the T-cell surface, which might sense both dopamine and CCL25 at the same time [23]. The transmembrane (TM) segments 5 (TM5) and TM6 from CCR9 and the TM5 and TM6 from DRD5 formed the interface of interaction required for the heteromer assembly [23]. Of note, similar to other heteromeric receptors described [24], the down-stream signalling pathways triggered by the stimulation of the CCR9:DRD5 complex are different from the signalling pathways triggered by the non-assembled DRD5 and CCR9. For instance, the dual stimulation of CCR9 and DRD5 in the assembled heteromer induces a reduction in cAMP levels without effects on the extent of ERK1/2 phosphorylation. Nevertheless, when the heteromer is disassembled, the same stimulation promotes increasing levels of cAMP and a high degree of ERK1/2 phosphorylation [23]. Thereby the heteromeric receptor represents an independent receptor triggering a unique biological effect different from the effects induced by the isolated forms of DRD5 and CCR9 [23]. This heteromeric complex is acquired by effector CD4^+^ T-cells as part of the gut tropism profile. Accordingly, the increase in CCL25 production and the reduction in dopamine levels associated with the colonic inflammation represent a dual signal stimulating the CCR9:DRD5 heteromer, which promotes the recruitment of CD4^+^ T-cells into the colonic lamina propria [23]. 

In this study, we aimed to analyse the molecular requirements involved in the heteromer assembly as well as to determine the underlying cellular mechanisms involved in the colonic tropism given by the stimulation of the CCR9:DRD5 complex. The results obtained here improve our understanding of the complex mechanisms involving heteromeric receptors associated with leukocyte migration in inflammation.

## 2. Results

### 2.1. The Dual Stimulation of the CCR9:DRD5 Heteromer on Primary CD4^+^ T-Cells Increases Their Migratory Rate in Confined Microenvironments

The migratory ability of leukocytes in 3D tissues is highly related to the migratory speed in confined microchannels [25,26]. To evaluate whether the gut tropism given by the CCR9:DRD5 heteromer to CD4^+^ T-cells is associated with a higher migratory ability induced by the dual stimulation of this heteromer, we conducted in vitro migration assays using confined microchannels. To this end, we first determined the migratory speed of primary CD4^+^ T-cells in 3 μm width microchannels in response to increasing concentrations of CCL25. The range of concentrations used in these experiments was chosen based on the range of CCL25 able to trigger a significant reduction in cAMP levels [23]. The results show that 50 ng/mL CCL25 already induced a significant increase in the migratory speed compared to the base migratory speed (Figure 1A). Next, we determined how was the migratory speed affected when dual stimulation of the heteromer was exerted. The results showed that either dopamine or the DRD5 agonist SKF81297 potentiated the migratory speed of primary CD4^+^ T-cells (Figure 1B). To address whether the speed potentiation observed in CD4^+^ T-cells when stimulated with CCR9 and DRD5 agonist was actually triggered by the CCR9:DRD5 heteromer, and not due to the individual stimulation of CCR9 and DRD5, we treated T-cells with a peptide able to disassemble the heteromeric receptor (TM6D) or with a control peptide (TM1D). Of note, these peptides were chosen on the basis of our previous results using a bimolecular complementation assay in which a T-cell line was transfected with the DRD5-nVenus and CCR9-cVenus fusion proteins, and then the ability of α-helix peptides analogue to TM segments from CCR9 and DRD5 to decrease the fluorescence associated with venus was determined [23]. The results show that, indeed, the speed potentiation exerted by CCL25 and SKF81297 was abrogated when the heteromer was disassembled (by TM6D), but not in the control situation (in the presence of TM1D) (Figure 1B). Thereby, these results represent causal evidence indicating that the dual simulation of the CCR9:DRD5 heteromer on CD4^+^ T-cells potentiates the migratory speed in confined microchannels.

### 2.2. The Dual Stimulation of the Heteromeric CCR9:DRD5 Complex Triggers the Activation of the Myosin II in Primary CD4^+^ T-Cells 

Myosin IIA has been described to play a key role in optimizing the motility rate of T-cells in 3D confined microenvironments in vivo and in vitro [25]. To evaluate whether the signalling triggered by the CCR9:DRD5 heteromer is related to the activation of Myosin IIA, we next evaluated the phosphorylation of myosin light chain 2 (pMLC2) in CD4^+^ T-cells under the dual stimulation of CCR9 and DRD5. Indeed, the stimulation of T-cells with CCL25 and SKF81927 induced a high degree of phosphorylation of this protein (Figure 2A,B). Importantly, the pre-treatment of T-cells with the TM6C peptide, which disrupted the assembly of CCR9 with DRD5, strongly reduced the extent of pMLC2. Conversely, the pre-treatment of CD4^+^ T-cells with the peptide TM7C, which is irrelevant for CCR9:DRD5 assembly, did not affect the degree of pMLC2 induced by CCL25 and SKF81927 (Figure 2A,B). Thus, these results indicate that the dual stimulation of the CCR9:DRD5 heteromer triggers the activation of myosin IIA, which is essential for optimizing the migration of T-cells in vivo.

### 2.3. The CCR9:DRD5 Heteromer Plays a Critical Role in the Recruitment of CD4^+^ T-Cells into the Colonic Mucosa upon Inflammation

Our previous study revealed a higher degree of lymphocytes expressing the CCR9:DRD5 heteromer in the inflamed colonic mucosa compared with healthy mucosa in human and animal models [23]. Due to the important role of the CCR9:DRD5 heteromer in triggering the activation of myosin IIA (Figure 2) and optimising the migratory ability of T-cells in confined microenvironments (Figure 1), we hypothesized that disassembling the CCR9:DRD5 heteromer should impair the recruitment of T-cells into the colonic mucosa upon inflammation in vivo. To address this possibility, we determined the infiltration rate of T-cells expressing the CCR9:DRD5 heteromer into the inflamed colonic mucosa when the heteromer formation was disrupted by the TM6C peptide compared with a control peptide that does not affect the heteromer assembly (TM7C). Accordingly, after imprinting gut tropism in CD4^+^ T-cells, they were treated with the experimental peptides (TM6C, TM7C, or vehicle) and then i.v. transferred into *Drd5^−/−^* recipient mice undergoing DSS-induced colitis. Three days later, the extent of heteromer^+^ cells was quantified in the colonic mucosa via in situ PLA assay. The results show a lower extent of heteromer^+^ T-cells reaching the colonic mucosa when the CCR9:DRD5 assembly was disrupted (using the TM6C peptide). However, the treatment of T-cells with a non-disrupting peptide (TM7C) did not affect the degree of heteromer^+^ T-cells reaching the colonic mucosa compared to T-cells treated with the vehicle (DMSO) (Figure 3). Of note, we confirmed that these peptides (see Appendix A) stayed in T-cells stably over time (Appendix A), did not kill the cells [23], and did not affect CCR9 expression (Appendix A) at the concentration used (4 μM). Thereby, together these results indicate that the assembled CCR9:DRD5 complex is required on CD4^+^ T-cells to lead to the infiltration of these cells into the colonic mucosa upon inflammation.

### 2.4. Analysing the Individual Contribution of Transmembrane Segments Involved in the Interface of the CCR9:DRD5 Heteromer to the Molecular Interaction

Afterwards, we studied the molecular interaction of CCR9 and DRD5 in silico. In our previous experiments using a bimolecular complementation assay in which a T-cell line was transfected with the DRD5-nVenus and CCR9-cVenus fusion proteins and then challenged with α-helix peptides analogue to TM segments from CCR9 and DRD5, we found that among the 14 peptides tested, only the TM5 and TM6 from CCR9 and TM5 and TM6 from DRD5 were able to inhibit the fluorescence associated with venus [23]. Based on these results, in the following experiments we studied the molecular interactions involved in these four TM segments, whilst TM1D and TM7C were used as control peptides. To explore a mechanistic model for the interaction between CCR9 and DRD5, we performed a series of non-equilibrium dynamic simulations to pull away specific TM segments of one protein from the other interacting protein. We measured the distance between the proteins and the force exerted during the simulations. Because the pulling force was applied in the form of an umbrella potential, the separation was not immediate. During the initial phase of the simulation, the force accumulated as it was opposed by the binding force between the intact protein and the TM being pulled away. At a certain point, the pulling force surpassed the intermolecular force, breaking the interactions between the proteins. We refer to this moment as the transition point, after which the TM was pulled away from the other protein linearly, following the rate defined in the simulation (5 nm/ns). The simulation continued until the force reached the defined value (1000 kJ/mol·nm^2^) and maintained a pseudo-equilibrium. To compare the behaviour of each TM being pulled away, we measured three parameters: the change in distance before the transition point, the time when the transition point was reached, and the maximum force exerted near the transition point (Appendix A). 

When DRD5 TMs were pulled away from the intact CCR9, TM5D and TM6D were more difficult to separate than the control TM1D, which is not experimentally involved in the interaction [23]. For TM5D and TM6D, the transition point was reached at later times, and the maximum force exerted was higher than the force needed to separate TM1D (Figure 4B,C). The change in distance before the transition point did not show a clear result, likely due to the instability of the binding of TM1D to CCR9, resulting in noisy data (Figure 4A). While unspecific interactions might explain this variability, it is evident that the interactions between TM5D/TM6D and CCR9 are stronger than those involving TM1D at the moment of transition.

When TMs from CCR9 were pulled away from DRD5, we also observed a clear difference between TM5C and TM6C and the control TM7C. The change in distance before transition was much greater for TM7C (Figure 5A), indicating weaker interactions. The transition point for TM7C was reached almost 100 ps earlier than for the other TMs (Figure 5B). Additionally, the maximum force exerted was higher for TM5C and TM6C compared to TM7C (Figure 5C). Altogether, these results obtained via in silico analysis suggest that TM5 and TM6 from CCR9 and DRD5 are involved in the interacting surface of the CCR9:DRD5 heteromer.

## 3. Discussion

It has been described that the acto-myosin cytoskeleton mediates force generation and protrusion during motility which is exerted by actin polymerization and class II myosin contraction of the actin network [27,28,29]. Non-muscle myosin IIA has been shown to play a critical role in regulating optimal T-cell ameboid motility [30]. This protein is the only class II myosin expressed in mouse T-cells and is regulated during lymphocyte arrest induced by TCR stimulation [31]. Moreover, myosin IIA activity also plays an important role in inducing uropodal detachment from highly adhesive surfaces such as ICAM-1-coated substrates [32,33]. Of note, the dynamics of myosin II are regulated by the phosphorylation of MLC2 [34]. Importantly, we observed here that the dual stimulation of the CCR9:DRD5 heteromeric complex triggers the phosphorylation of MLC2. Thereby, our results indicate that the CCR9:DRD5 heteromer is a key regulator of the migratory ability of T-cells.

CCR9 and DRD5 are GPCRs. Originally it was thought that GPCRs work as monomeric molecules; nevertheless, it is currently accepted that these receptors are generally found as homo- or hetero-oligomers [35]. Evidence has shown that heteromerization is selective: a particular GPCR can form heteromers just with some GPCR partners but not with others. Thus, during the past decade an increasing number of studies, performed mainly in the nervous system, reported heteromerization between different couples of GPCRs to form oligomers [36]. Compared with GPCR homomers, the assembly of GPCR heteromers leads to changes in the identity and affinity for agonist recognition, in the signalling coupled, and in the trafficking of participating receptors, thus strongly affecting their physiological function. An interesting example is the heteromer formed by the adenosine receptor A_2A_ (A_2A_R) and the DRD2, which can bind two different ligands. In this case, the binding of A_2A_R agonists to the A_2A_R:DRD2 heteromer induces a conformational change resulting in a reduction in DRD2 affinity for dopamine or DRD2-agonists and in a change in the coupling of DRD2 from Gi/o- to β-arrestin2-MAPK-mediated signalling [37]. Thus, this and many other examples in the literature illustrate how GPCRs heteromers involve complex interactions at the level of signalling and ligand affinity which can induce a final physiological outcome quantitatively and qualitatively different from those exerted by the corresponding protomers forming homomers.

With regard to the participation of DRD5 in the formation of heteromers, only DRD5:DRD1 [38] and DRD5:DRD2 [39,40] heteromers have been described so far. The DRD5:DRD1 heteromer is present in the kidney where, by coupling to the Gq-PLC-IP3 pathway, it plays a regulatory role in renal sodium transport [38]. On the other hand, DRD5:DRD2 heteromer stimulation involves the activation of the Gq-PLC-IP3 pathway in the nucleus accumbens, probably regulating acetylcholine release [39,40]. We recently described that DRD5 might form a heteromeric complex with CCR9, but not with CXCR4 [23]. We found that this heteromeric complex is acquired by CD4^+^ T-cells as part of the intestinal tropism profile, which allows these cells to be recruited into the colonic lamina propria upon inflammation [23].

In the context of inflammatory and autoimmune disorders, other GPCRs heteromers have been also described. For instance, CXCR3 and CXCR4, which are two important chemokine receptors involved in leukocyte migration involved in inflammation might form heteromers in which the stimulation of CXCR3 inhibits the binding of CXCR4 to its ligand [41]. Another example is the heteromer formed by the short-chain fatty acids receptors GPR41 and GPR43, which allow the communication between bacteria of the microbiota and host immune cells [42]. Notable, unlike the homomeric receptors GPR41 and GPR43, the GPR41:GPR43 heteromer lacks the ability to modify cAMP levels, but gains the ability to stimulate p38 phosphorylation, thus triggering a biological effect different from those induced by the stimulation of homomeric GPR41 and GPR43 [43]. These examples together with the present study illustrate how complex might be the molecular mechanisms undelaying inflammation. Further analysis of which GPCRs are expressed as heteromers and how these heteromeric modules behave in terms of signalling pathways would improve the understanding of the complex network of biological processes involved in autoimmune and inflammatory pathologies and would give a more clear overview for the design of therapies for treating these disorders. 

CCR9 and α4β7 have been previously described to be essential for the recruitment of T-cells upon colon inflammation [5,44]. For this reason, these homing molecules have taken the attention as molecular targets for the development of treatments for both CD and UC. Indeed, several drugs and humanized antibodies designed to break the CCR9-CCL25 and α4β7-MadCAM-1 interactions have been developed as therapeutic approaches for CD and UC [45,46]. However, these treatments attenuate the trafficking not only of effector T-cells, but also of Treg. Indeed, these molecules are necessary for the recruitment of Treg to the small intestine upon homeostatic conditions, a mechanism that involves the generation of oral tolerance [4]. This likely explains why vercirnon, a small molecule designed to inhibit CCR9 signalling, did not display significant beneficial effects for the treatment of CD [46]. This probably also explains why the treatment of CD patients with natalizumab, a monoclonal antibody that blocks the interaction of α4 integrin with its ligand, was accompanied by collateral effects such as the risk of developing progressive multifocal leukoencephalopathy [46]. It is noteworthy that a more selective targeting of leukocytes migration into the intestine has been reached with vedolizumab, a monoclonal antibody that blocks the action of α4β7, which does not present the potential collateral effects observed for natalizumab [46]. In the same line, the CCR9:DRD5 heteromer seems to be exclusively involved in the infiltration of CD4^+^ T-cells in the colonic mucosa upon inflammation, and not in homeostatic conditions, as *Drd5^−/−^* mice do not develop issues associated with food allergy [23].

Regarding the molecular characterisation of the interacting interphase of the CCR9:DRD5 heteromer, our in silico approach used here indicates that TM5 and TM6 from CCR9 and DRD5 are involved in the interacting surface of the CCR9:DRD5 heteromer. These results agree with our previous results obtained with a battery of peptides analogue to the TM segments from CCR9 and DRD5 where the disruption of the CCR9:DRD5 assembly was determined using a bimolecular complementation assay [23]. 

Importantly, our results show that disrupting the CCR9:DRD5 assembly induced a sharp reduction in the pMLC2 in vitro and a marked inhibition in the recruitment of CD4^+^ T-cells into the colonic mucosa in mice undergoing inflammatory colitis. Thereby, disassembling the formation of the CCR9:DRD5 represents an important therapeutic opportunity to treat IBD. It is important to note that our study does not rule out the possibility that the CCR9:DRD5 heteromer is involved in the regulation of another biological process in the body, an issue that should be addressed in future works. Another limitation of the present work is that our analysis does not rule out that some extracellular or intracellular loops of CCR9 and DRD5 might also contribute to the interface of interaction required for the heteromer assembly. Despite this study and our previous work [23] which bring together in vivo, in vitro, and in silico evidence of the relevance of the CCR9:DRD5 heteromer in T-cells recruitment into the colon upon inflammatory conditions, the therapeutic potential should be tested in vivo using preclinical models of IBD in the future. These findings encourage the design of strategies targeting this heteromeric complex as treatment of IBD, such as dual antagonists directed to affect both protomers in the heteromer, heteromer-selective drugs (drugs that bind a GPCR with higher affinity when it is part of a heteromer), and non-digestible drugs (i.e., small molecules) or monoclonal antibodies avoiding the heteromer assembly.

## 4. Materials and Methods

### 4.1. Mice

Wild-type C57BL/6 (WT; *Drd5^+/+^*) mice were obtained from The Jackson Laboratory. C57BL/6 *Drd5^−/−^* mice were kindly donated by Dr. David Sibley [47]. Mice aged from 6 to 10 weeks were used in all experiments. All procedures performed on animals were approved by and complied with regulations of the Institutional Animal Care and Use Committee at Fundación Ciencia & Vida.

### 4.2. Reagents

Monoclonal antibodies (mAbs) for flow cytometry: anti-α4β7 (clone DATK32) conjugated to PE and anti-CCR9 (clone CW.1.2) conjugated to AlloPC or to AlloPC-Cy7 were obtained from eBioscience (San Diego, CA, USA). Anti-CD4 (clone GK1.5) conjugated to AlloPC and AlloPC-Cy7; anti-CD25 (clone PC61) conjugated to Fluorescein isothiocyanate (FITC); anti-CD44 (clone IM7) conjugated to PE; anti-CD62L (clone MEL14) conjugated to AlloPC-Cy7; and anti-TCR β chain (clone H57-597) conjugated with PerCP/Cy5.5 were purchased from Biolegends (San Diego, CA, USA). mAbs for cell culture: the following mAbs low in endotoxins and azide free were purchased from Biolegend: anti-CD28 (clone 37.51) and anti-CD3ε (clone 145-2C11). The rabbit anti-phospho Ser19 MLC2 mAb (pMLC2; clone 3671) was obtained from Cell Signaling Technology (Beverly, MA, USA). Carrier-Free CCL25 was purchased from Biolegend. IL-2 was obtained from PreproTech (Rocky Hill, NJ, USA). Zombie Aqua (ZAq) Fixable Viability dye detectable via flow cytometry was purchased from Biolegend. Retinoic acid (RA) was purchased from Sigma-Aldrich (San Luis, MO, USA). Fetal Bovine Serum (FBS) was obtained from Life Technologies (Carlsbad, CA, USA). The peptide analogues to transmembrane (TM) segments derived from CCR9 and DRD5 (Appendix A) were synthesized by GenScript (Piscataway, NJ, USA). Anti-CD3/anti-CD28-conjugated dynabeads were purchased from Thermo Scientific. Bovine Serum Albumin (BSA) was purchased from Rockland (Limerick, PA, USA). Dextran Sodium Sulphate (DSS) was obtained from TdB Labs (Uppsala, Sweden). All tissue culture related reagents were bought from Life Technologies. SKF81297 was obtained from Tocris (Bristol, UK).

### 4.3. Flow Cytometry Analysis of CD4^+^ T-Cell Phenotypes

Cells were stained with ZAq Fixable Viability kit, followed by fluorochrome-conjugated mAbs specific to cell-surface markers in PBS containing 2% FBS for 30 min. The surface markers analysed included α4β7, CCR9, CD3, CD4, CD25, CD44, CD62L, and TCRβ. Afterwards, cells were fixed with 1% paraformaldehyde in phosphate-buffered saline (PBS, Na_2_HPO_4_ 8.1 µM, KH_2_PO_4_ 1.47 µM, NaCl 64.2 mM, KCl 2.68 mM, pH 7.4) for 15 min at room temperature, washed twice with PBS, and analysed in a flow cytometer. To analyse the phosphorylation of MLC2, after the immunostaining of surface markers, cells were fixed and permeabilised using the Foxp3 Fix/Perm buffer (Bioleged) instead of fixing with paraformaldehyde. Cells were then incubated with the rabbit anti-pMLC2 mAb (1:100) at 4 °C for 1 h, followed by incubation with FITC-conjugated anti-rabbit IgG Abs (Santa Cruz Biotechnology, Santa Cruz, CA, USA) at 4 °C for 1 h. Non-specific rabbit Ig followed by the secondary FITC-conjugated anti-rabbit IgG Ab were used as controls. Data were collected with a Canto II (BD), and results were analysed with FACSDiva (BD) and FlowJo v9/X software (Tree Star, Ashlan, OR, USA). FMO was used to control the detection of each parameter in flow cytometry.

### 4.4. Imprinting Gut Tropism in CD4^+^ T-Cells Ex Vivo

Naïve (CD3^+^ CD4^+^ CD44^−^ CD62L^+^) T-cells were isolated from the spleen of WT (*Drd5^+/+^*) mice via cell sorting using a FACS Aria II (BD), obtaining purities over 98%. Gut tropism was imprinted via activation of T-cells in the presence of RA and IL-2, as described before [48]. Briefly, naïve T-cells were resuspended (10^6^ cells/mL) in RPMI1640 medium containing 10% FBS, 2 mM L-glutamine, 1% Penicillin/Streptomycin, MEM Non-Essential Amino Acids 1X and Sodium Pyruvate 1X, gentamicin 50 µg/mL and β-mercaptoethanol 1 µg/mL. Cells were activated with anti-CD3/CD28-coated dynabeads at a beads/cells ratio of 1:1 in the presence of 100 nM all-trans RA and 1000 U/mL recombinant mouse IL-2 for 5 days. Viability and gut tropism were routinely confirmed after 5 days of culture by staining with ZAq Fixable Viability kit and CCR9 and α4β7 immunostaining followed by flow cytometry analysis.

### 4.5. T-Cell Migration in Microchannels

Naïve CD4^+^ T-cells were isolated from the spleen of WT mice and incubated in conditions to induce gut tropism (see above). CCR9^+^ cells were purified from CD4^+^ T-cells bearing gut tropism using a cell-sorter. Lymphocytes were incubated with or without TM analogue peptides (4 μM) for 4 h and then the median speed of cells was determined in microchannels as described before [49]. Briefly, 5 μL of cell suspension (10^7^ cells/mL) was loaded into a fibronectin (10 µg/mL)-coated chip of poly-dimethylsiloxane containing several 3 μm-diameter micro-channels. After 30 min of incubation at 37 °C, 90% humidity, and 5% CO_2_, 2 mL of complete culture medium containing IL-2 (100 U/mL), CCL25 (0–200 ng/mL), and dopamine (0–1 μM) or SKF81297 (0–1 μM) were added. Afterwards, cell phase contrast images were recorded during 10 h with 8 min time-lapses using an automated microscope (Nikon ECLIPSE TE1000-E (Tokyo, Japan), and Olympus X71 (Tokyo, Japan), with a Marzhauser motorized stage and an HQ2 Roper camera) equipped with an environmental chamber to control temperature (37 °C), humidity (90%), and CO_2_ (Life Imaging Services, Basel, Switzerland). The analysis of migration parameters was performed using an ImageJ (v1.54f) Fiji-based script. Base speed was calculated in every experiment to control whether positive or negative effects were exerted on lymphocyte migration.

### 4.6. Dextran Sodium Sulphate-Induced Acute Inflammatory Colitis

*Drd5^−/−^* mice were treated with 1.75% DSS in the drinking water for six days. Three days after the beginning of DSS treatment, mice received an i.v. injection of *Drd5^+/+^* CD4^+^ T-cells (6 × 10^6^ total cells per mouse). Seventy-two hours later, mice were sacrificed, and colonic tissue was collected for further analysis. 

### 4.7. In Situ Proximity Ligation Assay 

Colonic sections of mice undergoing inflammatory colitis were used to analyse the CCR9:DRD5 heteromer in situ via proximity ligation assay (PLA). Tissue sections were fixed in 4% paraformaldehyde for 15 min, washed with PBS containing 20 mM glycine to quench the aldehyde groups and permeabilized with the same buffer containing 0.05% Triton X-100 for 15 min. Primary antibodies recognising CCR9 (rabbit anti-CCR9; 1:100 dilution; purchased from Invitrogen, Waltham, MA, USA) and DRD5 (mouse mAb anti-DRD5; 1:100 dilution; purchased from Invitrogen) were used with the Duolink In Situ Red Starter Kit Mouse/Rabbit (Sigma-Aldrich, St. Louis, MI, USA) according to the manufacturer instructions. Nuclei were stained with Hoechst (1:200 dilution; purchased from Sigma-Aldrich). Coverslips were mounted using mowiol solution. Samples were observed in a Leica SP2 confocal microscope (Leica Microsystems, Mannheim, Germany) equipped with an apochromatic 63× oil-immersion objective (N.A. 1.4) and 405 nm and 561 nm laser lines. For each field of view a stack of two channels (one per staining) and 3 to 4 Z stacks with a step size of 1 µm were acquired. Quantification of cells containing one or more red spots versus total cells (blue nucleus) was determined using the ImageJ software (from the National Institute of Health, Bethesda, MD, USA). In all experiments, a group of mice received the injection of PBS instead of lymphocytes, which was used to control the limits of detection and quantification of PLA+ cells in the colonic mucosa.

### 4.8. Protein Modelling 

The sequence of murine proteins were extracted from Uniprot (Q8BLD9 for DRD5; Q9WUT7 for CCR9). Structures were obtained from modelling, using Modeller v10 [50]. For murine CCR9, the human CCR9 structure was used as a template (5LWE [51]), which allowed for the modelling of the segment 32-344. Since extracellular loop 2 (ECL2) was not resolved in the template structure, we used the ECL2 sequence obtained from human proteins CXCR4 (3ODU [52]) and CCR5 (4MBS [53]) as templates for modelling this loop, as these proteins display high sequence homology with CCR9 when analysed with BLASTp server available at the NCBI website (https://blast.ncbi.nlm.nih.gov/blast/Blast.cgi, access on 04 September 2024). These templates were selected only based on their sequence similarity with CCR9. Analogously, based only on their sequence similarity with DRD5 determined via BLASTp, we used *Meleagris gallopavo* β1 adrenergic receptor structures as templates for modelling DRD5. Since ELC2 and ILC3 were not resolved in the *Meleagris gallopavo* β1 adrenergic receptor structures, we combined several available structures (2Y00 [54]; 2VT4 [55]; 5A8E [56]; and 4BVN [57]) and implemented the information defining the boundaries of all TM segments available in the GPCRdb [58], which allowed for the modelling of the segment 30-369 (including ECL2 and ICL3).

### 4.9. Heterodimer Modelling 

Previous results had shown that TM5 and TM6 of both G protein-coupled receptors (GPCRs) were involved in the interaction surface. Structural alignment with the experimental structure of the μ-opioid receptor dimer (4DKL [59]) was used to identify possible interacting residues between DRD5 and CCR9, because this dimer also interacts through their TM5 and TM6. The selected residues were used to feed the HADDOCK2.2 server [60], as experimental restraints, and then 10,000 heterodimer models were obtained. Using HADDOCK filters and visual inspection, a final DRD5-CCR9 dimer model was chosen.

### 4.10. Equilibrium Simulation 

The DRD5:CCR9 complex was uploaded to the CHARMM-GUI server [61] to embed it in a POPC (palmitoyl-oleoyl phosphatidylcholine) bilayer and a 17 × 17 × 14 nm box of solvent (TIP3P water and 150 mM NaCl). The recommended CHARMM-GUI protocol was used to relax the system using molecular dynamics. The program GROMACS 2022 [62] was used along with the CHARMM36 forcefield [63] to describe the system. Roughly, it consists of adding a position restraint for specific atoms and then gradually reducing these restrictions until they become zero. In all simulations, a timestep of 2 femtoseconds was used, and PME (Particle-Mesh Ewald) was used to calculate long-range electrostatic interactions. In the final steps of equilibration, an NPT ensemble was used at 310.15 K and 1 atm of temperature and pressure, respectively. Finally, a production simulation was performed without any restriction for 500 ns at the NPT ensemble.

### 4.11. Non-Equilibrium Simulation 

The final structure from the production simulation served as a template for building several new structures, where one protein remained intact and the other was trimmed to include only one TM segment. Six structures were constructed: CCR9–TM1D, CCR9–TM5D, CCR9–TM6D, DRD5–TM5C, DRD5–TM6C, and DRD5–TM7C. TM5C/D and TM6C/D form the interface of the heterodimer, while TM1D and TM7C were used as controls. Please note that sequences for TM segments used for in silico analyses are the same sequences for peptides shown in Appendix A but without the TAT peptide (shown in red) and without changing cysteines with serines (shown in green). These new protein systems were re-embedded in a similar bilayer and solvent box as before, in order to accommodate lipids surrounding the TM segments, creating a new 14 × 23 × 14 nm box using the InflateGRO methodology [64]. Non-equilibrium pulling simulations were performed using GROMACS 2022 with an umbrella pulling protocol. The TM segment was pulled away from its protein partner, which was held in place with position restraints on the main chain atoms. The simulation ran for 750,000 steps with a pull rate of 5 nm/ns and a force constant of 1000 kJ/mol·nm^2^.

### 4.12. Statistical Analysis

Comparison between multiple groups was analysed using 1- or 2-way ANOVA with peptide treatment or ligand treatment as the independent factor. When ANOVA showed significant differences, pair-wise comparison between means was tested with Tukey’s or Sidak’s post hoc test. *p* value ≤ 0.05 was considered significant. Analyses were performed with GraphPad Prism 9 software. 

## 5. Conclusions

In this study, we demonstrated that the CCR9:DRD5 heteromeric complex plays a relevant role in the migration of CD4^+^ T-cells into the colonic mucosa upon inflammation. The mechanistic analyses show that dual stimulation of this heteromeric complex triggers the phosphorylation of MLC2, which is essential for acquiring an optimal migration in confined microenvironments.

## 6. Patents

R.P., F.O-B., and J.C. are named inventors on a pending patent application describing the therapeutic use of novel peptides that decrease the homing of CD4^+^ T-cells with gut tropism by blocking the assembly of dopamine receptor D5 and C-C chemokine receptor 9. This is intended as a treatment for inflammatory bowel diseases and could be construed as a potential conflict of interest.

## Figures and Tables

**Figure 1 ijms-25-10022-f001:**
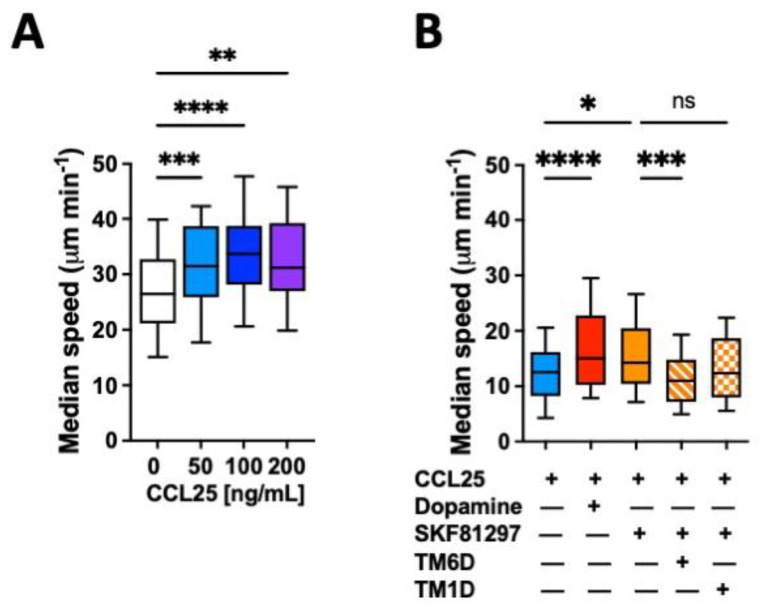
CCR9:DRD5 heteromer signalling increases the migratory speed of CD4^+^ T-cells in microchannels. Naïve CD4^+^ T-cells were isolated from the spleen of wild-type mice (*Drd5^+/+^*) and then activated with anti-CD3/anti-CD28 mAbs-coated dynabeads in the presence of IL-2 and RA for 5 d to induce gut tropism. Afterwards, cells were individually tracked in 3 μm-width confined microchannels under different conditions and the migratory speed was determined. (**A**) Migratory speed in response to increasing concentrations of CCL25 was determined; *n* = 52–163 cells per condition. (**B**) Migratory speed was determined in the presence of CCL25 (50 ng/mL) alone or with dopamine (1 μM), or with SKF81297 (100 nM) in the absence or in the presence of a TM-peptide irrelevant for CCR9:DRD5 heteromer assembly (TM1D) or of a TM-peptide that disrupt CCR9:DRD5 heteromer assembly (TM6D); *n* = 83–160 cells per condition. (**A**,**B**) Data correspond to the median migratory speed of lymphocytes in μm/min. In the box plots, the bars include 90% of the data points, the horizontal line in the box indicates the median and the box contains 75% of the data points. Data from two independent experiments are shown. *, *p* < 0.05; **, *p* < 0.01; ***, *p* < 0.001; ****, *p* < 0.0001 via one-way ANOVA followed by Tukey’s post hoc test. ns, non-significant.

**Figure 2 ijms-25-10022-f002:**
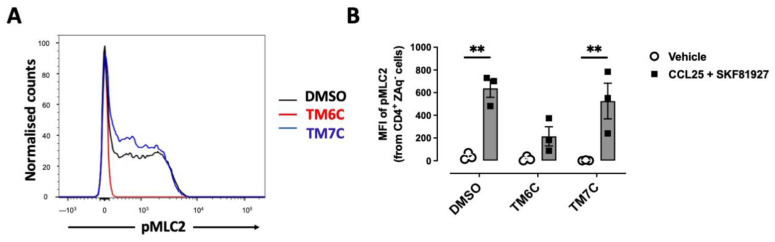
CCR9:DRD5 heteromer signalling involves the activation of the myosine light chain 2. Naïve CD4^+^ T-cells were isolated from the spleen of wild-type mice (*Drd5^+/+^*) and then activated with anti-CD3/anti-CD28 mAbs-coated dynabeads in the presence of IL-2 and RA for 5 d to induce gut tropism. During the last 4 h, cells were non-stimulated (vehicle) or stimulated with CCL25 (300 ng/mL) and SKF81297 (100 nM) in the presence of 4 μM of peptides TM6C or TM7C, or only DMSO as a control. Cells were stained for extracellular expression of CD4 and intracellular phosphorylation of MLC2. (**A**) Representative histograms showing the fluorescence distribution associated with the immunostaining of pMLC2 in the CD4^+^ live (ZAq^−^) population. (**B**) Quantification of the extent of pMLC2. Values are the mean fluorescence intensity associated with the immunostaining of pMLC2. Data are represented as the mean ± SEM from three independent experiments. **, *p* < 0.01 via two-way ANOVA followed by the Tukey’s multiple comparisons post hoc test.

**Figure 3 ijms-25-10022-f003:**
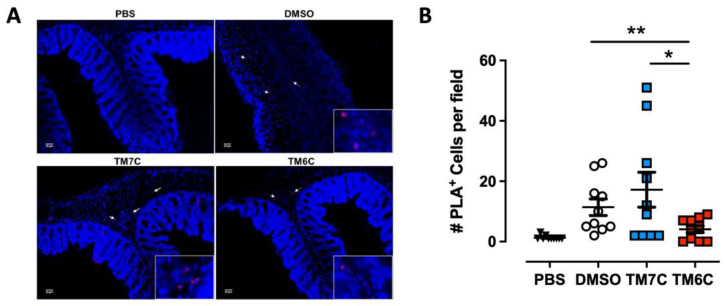
The disassembling of the CCR9:DRD5 heteromer reduces the recruitment of CD4+ T-cells into the colonic mucosa. Naïve CD4^+^ T-cells were isolated from the spleen of wild-type mice (*Drd5^+/+^*) and then activated with anti-CD3/anti-CD28 mAbs-coated dynabeads in the presence of IL-2 and RA for 5 d to induce gut tropism. During the last 4 h, cells were treated with 4 μM of peptides TM6C or TM7C, or only DMSO as a control. Afterwards, cells were i.v. transferred (6 × 10^6^ cells/mouse) into *Drd5^−/−^* mice which were previously exposed to 1.75% DSS for 3 d. Mice were exposed to 1.75% DSS for 3 more days, sacrificed, and PLA was conducted on colonic tissue. (**A**) Representative images obtained from PLA analysis. Arrows show PLA^+^ cells. Bar, 20 μm. Images with higher magnification are shown in the right-bottom corner. (**B**) Quantification of PLA^+^ cells. Values are the number of PLA^+^ cells per field. Data are represented as the mean ± SEM from three independent experiments. Each symbol represents a different field. *, *p* < 0.05; **, *p* < 0.01 via one-way ANOVA followed by Sidak’s multiple comparisons post hoc test.

**Figure 4 ijms-25-10022-f004:**
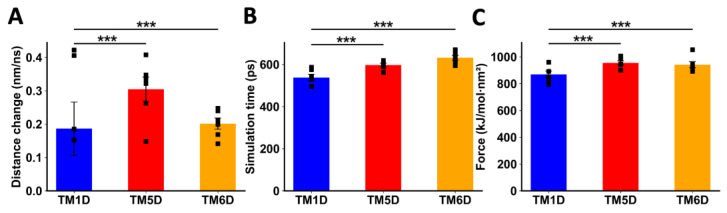
Analysis of non-equilibrium simulation during the pulling away of DRD5 TMs from the whole CCR9 protein. (**A**) Distance change during simulation at the first part of the method, before the transition point. (**B**) Simulation time when the transition point is reached. (**C**) Maximum force reached around transition point. Each symbol represents an individual simulation (*n* = 6). The mean ± SEM is shown. ***, *p* < 0.001 via one-way ANOVA followed by Sidak’s multiple comparisons post hoc test.

**Figure 5 ijms-25-10022-f005:**
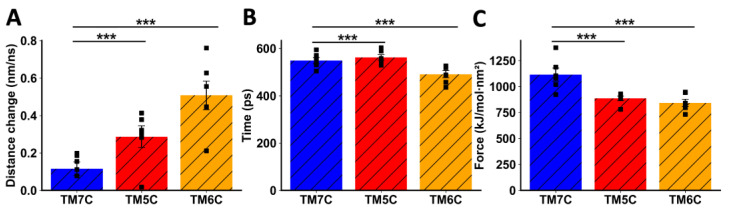
Analysis of non-equilibrium simulation during the pulling away of CCR9 TMs from the whole DRD5 protein. (**A**) Distance change during simulation at the first part of the method, before the transition point. (**B**) Simulation time when the transition point is reached. (**C**) Maximum force reached around the transition point. Each symbol represents an individual simulation (*n* = 6). The mean ± SEM is shown. ***, *p* < 0.001 via one-way ANOVA followed by Sidak’s multiple comparisons post hoc test.

## Data Availability

Data are contained within this article.

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
