# Peer review of "Chemokinergic and Dopaminergic Signalling Collaborates through the Heteromer Formed by CCR9 and Dopamine Receptor D5 Increasing the Migratory Speed of Effector CD4+ T-Cells to Infiltrate the Colonic Mucosa"

_ijms, 2024, doi:10.3390/ijms251810022_

Round 1

Reviewer 1 Report

Comments and Suggestions for Authors

The manuscript titled "Chemokinergic and Dopaminergic Signalling Collaborates through the Heteromer Formed by CCR9 and Dopamine Receptor D5 Increasing the Migratory Speed of Effector CD4+ T-Cells to Infiltrate the Colonic Mucosa" presents a compelling investigation into the role of CCR9 heteromers in T-cell migration during inflammatory bowel disease (IBD). The study is well-conceived and provides valuable insights into the molecular mechanisms of T-cell recruitment to the colonic mucosa. The background information is sufficient for understanding the study's rationale, and the experimental design is robust, supporting the authors' conclusions. However, there are areas where additional detail and clarification would enhance the manuscript. Such specific section-wise points for revision are outlined in the attached file

Author Response

REVIEWER 1

The manuscript titled "Chemokinergic and Dopaminergic Signalling Collaborates through the Heteromer Formed by CCR9 and Dopamine Receptor D5 Increasing the Migratory Speed of Effector CD4+ T-Cells to Infiltrate the Colonic Mucosa" presents a compelling investigation into the role of CCR9 heteromers in T-cell migration during inflammatory bowel disease (IBD). The study is well-conceived and provides valuable insights into the molecular mechanisms of T-cell recruitment to the colonic mucosa. The background information is sufficient for understanding the study's rationale, and the experimental design is robust, supporting the authors' conclusions. However, there are areas where additional detail and clarification would enhance the manuscript. Such specific section-wise points for revision are outlined below.

Q1. In the introduction section, the authors outline the prerequisites for assembling the CCR9 heteromeric complex on CD4+ T-cells. Could the authors elaborate further on the specific molecular interactions and structural requirements essential for this assembly? Additionally, if there are existing studies that identify known cofactors or post-translational modifications required for the formation of the CCR9 heteromeric complex. If yes, could these be cited and discussed in the manuscript? Including this information would enhance the manuscript's clarity and accessibility for readers.

ANSWER: We thank the reviewer for this observation. We have included further detail about it in the introduction: “The transmembrane (TM) segments 5 (TM5) and TM6 from CCR9 and the TM5 and TM6 from DRD5 formed the interface of interaction required for the heteromer assembly (23)” (lines 79-80).

Q2. The authors mentioned that the signaling pathways triggered by the CCR9 complex differ from those initiated by the non-assembled forms of DRD5 and CCR9. Could you elaborate on these differences and how they contribute to the unique biological effects observed? This would help make the distinction more precise and more succinct.

ANSWER: We have included a more precise explanation for this in the introduction: “For instance, the dual stimulation of CCR9 and DRD5 in the assembled heteromer induces a reduction of cAMP levels without effects on the extent of ERK1/2 phosphorylation. Nevertheless, when the heteromer is disassembled, the same stimulation promotes increasing levels of cAMP and a high degree of ERK1/2 phosphorylation (23).” (lines 83-87).

Q3. The current study describes a switch in dopamine receptor activation from DRD1/DRD2 to DRD3/DRD5 under inflammatory conditions. Could the authors provide a clearer explanation of how this switching occurs at the molecular level? It becomes necessary if you are mentioning switching of receptor activation. Additionally, the study has mentioned the involvement of these receptors in inflammation, could the authors describe the specific contributions of DRD3 and DRD5 in modulating immune responses, exclusively in relation to their role in the CCR9 heteromer? Providing this detail will help readers understand receptor’s precise role in inflammatory reactions.

ANSWER: We thank the reviewer for this comment. Indeed, the molecular mechanism underlying has not been explored. We included a sentence to clarify this issue in the third paragraph of the introduction: “Under homeostatic conditions, dopamine is found at high concentrations in the colonic mucosa in humans (8) and mice (9,10), which might stimulate both high-affinity (DRD3 and DRD5) and low-affinity (DRD1 and DRD2) dopamine receptors. Although the signalling of DRD1 and DRD2 is dominant over DRD3 and DRD5 at high levels of dopamine, how it happens at the molecular level is not understood” (lines 60-64).

We also included more details about the effect of DRD3-signalling on CCR9 upon inflammation “Interestingly, the stimulation of DRD3 in Treg also down-regulates the CCR9 expression, thus impairing the recruitment of Treg to the colonic mucosa and exacerbating the inflammatory process (18)” (lines 72-75). The detailed effect of DRD5-signalling on CCR9 upon inflammation is indeed the whole fourth paragraph in the introduction. 

Q4. Could the authors clarify the criteria used for selecting the specific templates for modeling murine CCR9 and DRD5? Were factors beyond structural similarity, such as functional relevance or the resolution of the template structures, considered in this selection process?

ANSWER: Only the sequence similarity was considered in the selection of template structures for modeling both proteins, structural or functional considerations were not considered. The section 4.8 has been rewritten to improve the clarify about it.

Q5. Could the authors clarify how the decision was made to use human CXCR4 and CCR5 for modeling the ECL2 of murine CCR9? Additionally, could they explain the rationale for selecting various Meleagris gallopavo β1 adrenergic receptor structures for modeling the ECL2 and ICL3 of murine DRD5? Including these details is essential for ensuring the manuscript's clarity and coherence.

ANSWER: We thank the reviewer for raising this point. We have rewritten the section 4.8 to improve the clarity of these points in the new version of the paper.

Q6.The authors observed noteworthy differences in the behavior of TM5C, TM6C, and TM7C during their analysis, indicating weaker interactions for TM7C and a transition point reached earlier than for the other TMs. Moreover, the force exerted was higher for TM5C and TM6C compared to TM7C. It was noted that TM5 and TM6 of both GPCRs were involved in the interaction surface, and previous results indicated that TM5 and TM6 also mediate interactions in the DRD5-CCR9 dimer. Can the authors provide more details on whether other transmembrane regions were considered for potential interactions in the DRD5-CCR9 dimer, or if the focus was solely on TM5 and TM6? Please specify the reasoning behind these conclusions.

Q6- old Based on the authors' mention that TM5 and TM6 of both G protein-coupled receptors (GPCRs) were involved in the interaction surface and that the DRD5 and CCR9 dimer also interacts through TM5 and TM6, could the authors provide more detail on whether other transmembrane regions were considered for potential interactions in the DRD5-CCR9 dimer? If so, could they explain why these regions were ultimately excluded, and if there were any drawbacks associated with including other segments or regions? Please specify the reasoning behind these conclusions.

ANSWER: We have included more details to clarify this selection of peptides.

At the beginning of section 2.4: “Our previous experiments using bimolecular complementation assay in which a T cell line was transfected with the DRD5-nVenus and CCR9-cVenus fusion proteins and then challenged with a-helix peptides analogue to TM segments from CCR9 and DRD5, we found that among the 14 peptides tested, only the TM5 and TM6 from CCR9 and TM5 and TM6 form DRD5 were able to inhibit the fluorescence associated to venus (23). Based on these results, in the following experiments we studied the molecular interactions involved in these four TM segments, whilst TM1D and TM7C were used as control peptides” (lines 232-239).

At the introduction: “The transmembrane (TM) segments 5 (TM5) and TM6 from CCR9 and the TM5 and TM6 from DRD5 formed the interface of interaction required for the heteromer assembly (23)” (lines 79-80).

Q7. In their result section (Figure 1A), the authors have shown a significant increase in migratory speed with 50 ng/mL CCL25. Did you test concentrations of CCL25 both higher and lower than this value? If so, how did these different concentrations affect migratory speed? Additionally, is there a dose-response curve available that can be shared? Please specify these doubts as clarifying these points will help in understanding the manuscript thoroughly.

ANSWER: The results shown in the figure 1A indeed correspond to a dose-response curve of migratory speed in response to increasing concentrations of CCL25. The range of concentrations used in these experiments was chosen based on the range of CCL25 able to trigger significant reduction of cAMP levels in our previous study (reference 23).

This has been clarified in the first paragraph of the section 2.1: “The range of concentrations used in these experiments was chosen based on the range of CCL25 able to trigger significant reduction of cAMP levels (23)” (Lines 128-129).

Q8. The authors have used TM1D and TM7C as a control peptides, could they stipulate more details on the control experiments conducted to assess the specificity of the interactions? Specifically, how were the controls TM1D and TM7C used to validate the findings for TM5D and TM6D? Additionally, were there any other controls or reference points included in the simulations to ensure the robustness of the results?

ANSWER: We have added a sentence in the first paragraph of section 2.1. to include more details of how these peptides were chosen, thus improving the clarity: “Of note, these peptides were chosen on base of our previous results using a bimolecular complementation assay in which a T cell line was transfected with the DRD5-nVenus and CCR9-cVenus fusion proteins and then the ability of a-helix peptides analogue to TM segments from CCR9 and DRD5 to decrease the fluorescence associated with venus was determined (23)” (Lines 138-142).

Q9. In the Section 2.3, the authors have mentioned that the peptides used to disrupt the heteromer were stable over time and did not affect CCR9 expression. Did they assess the potential off-target effects or any toxicity of these peptides in vivo or in vitro beyond CCR9 expression? How do you confirm that the observed effects are specifically due to the disassembly of the CCR9 heteromer rather than other unintended interactions?

ANSWER: In our previous study, we determined whether the treatment with the different peptides was potentially toxic affecting the viability of T-cells, and we observed no significant changes in the extent of cell death compared to cells that were no exposed to peptides. This is indicated at the end of the section 2.3 in the paper “Of note, we confirmed that these peptides (see table S1) stayed in T-cells stably in the time (Fig. S1), did not kill the cells (22), and did not affect CCR9 expression (Fig. S2) at the concentration used (4 mM)” (lines 210-212).

Q10. In the discussion sections the authors have emphasized the importance of GPCR heteromerization in modulating receptor function. Do you anticipate that similar heteromeric complexes might play a role in other inflammatory or autoimmune diseases? If so, how might your results be regarding CCR9 effect future research in these areas? Can you add few lines in regarding this? This can offer a broader implication for GPCR heterodimers.

ANSWER: We have included a new paragraph (fourth paragraph) about this topic in the discussion: “In the context of inflammatory and autoimmune disorders, other GPCRs have been also described. For instance, CXCR3 and CXCR4, which are two important chemokine receptors involved in leukocyte migration involved in inflammation might form heteromers in which the stimulation of CXCR3 inhibits the binding of CXCR4 to its ligand (41). Another example is the heteromer formed by the short-chain fatty acids receptors GPR41 and GPR43, which allow the communication between bacteria of the microbiota and host immune cells (42). Notable, unlike the homemeric receptors GPR41 and GPR43, the GPR41:GPR43 heteromer lacks the ability to modify cAMP levels, but gains the ability to stimulate p38 phosphorylation, thus triggering a biological effect different of those induced by the stimulation of homemeric GPR41 and GPR43 (43). These examples together with the present study illustrate how complex might be the molecular mechanisms undelaying inflammation. Further analysis of which GPCRs are expressed as heteromers and how these heteromeric modules behave in terms of signalling pathways would improve the understanding of the complex network of biological processes involved in autoimmune and inflammatory pathologies, and would give a more clear overview for the design of therapies for treating these disorders”.

Q11. Additionally, the discussion section underlines the therapeutic potential of targeting the CCR9 heteromer in inflammatory bowel disease (IBD). Could the authors make few points to compare the potential benefits of this approach with existing therapies? Specifically, how might targeting the CCR9 heteromer provide more selective immune modulation compared to current treatments that target CCR9 and α4β7 integrin? Including a few lines of comparison of these aspects could enhance the readability and appeal of this section.

ANSWER: We have expanded the discussion about the potential advantages of therapeutic approaches targeting more selectivity the T-cell infiltration into the gut by disassembling the CCR9:DRD5 heteromer in the fifth paragraph of the discussion: “This is probably the reason of why vercirnon, a small molecule designed to inhibit CCR9 signalling, did not display significant beneficial effects for the treatment of CD (46). This likely also the reason of why the treatment of CD patients with natalizumab, a monoclonal antibody that blocks the interaction of a4 integrin with its ligand, was accompanied with collateral effects such as the risk of developing progressive multifocal leukoencephalopathy (46). It is noteworthy that a more selective targeting of leukocytes migration into the intestine has been reached with vedolizumab, a monoclonal antibody that blocks the action of α4β7, which does not present the potential collateral effects observed for natalizumab (46). In the same line, the CCR9:DRD5 heteromer seems to be exclusively involved in the infiltration of CD4+T-cells in the colonic mucosa upon inflammation, and not in homeostatic conditions, as Drd5-/- mice do not develop issues associated with food allergy (23)” (lines 347-358).

Suggestion- While the in vitro and in silico results are promising, the discussion could benefit from explicitly addressing the need for further in vivo validation using more complex models or eventual clinical trials. Highlighting these next steps and how the findings might be translated into clinical settings could provide a clearer path forward for future research. This is just a suggestion if authors could add few lines making these points in the discussion section that can make it more approachable and authentic.

ANSWER: As suggested by the reviewer, we have included some lines at the end of the discussion about this point: “Despite this study and our previous work [23] bring together in vivo, in vitro and in silico evidence of the relevance of the CCR9:DRD5 heteromer in T-cells recruitment into the colon upon inflammatory conditions, the therapeutic potential should be tested in vivo using preclinical models of IBD in the future. These findings encourage the design of strategies targeting this heteromeric complex as treatment of IBD, such as dual antagonists directed to affect both protomers in the heteromer, heteromer selective drugs (drugs that bind a GPCR with higher affinity when it is part of an heteromer), and non-digestible drugs (i.e. small molecules) or monoclonal antibodies avoiding the heteromer assembly” (lines 375-383).

Reviewer 2 Report

Comments and Suggestions for Authors

This article targets the role of the heteromeric CCR9:DRD5 complex in the migration of CD4+ T cells in the colonic mucosa upon inflammation, thus raising the therapeutic opportunity for treating IBD. The topic is relevant, but major deficiencies in form and content need to be addressed:

1. The similarity ratio shows a very high percentage (40%) of similarity with other works including the authors' own work. It is imperative that this percentage be significantly reduced, especially since it is an original research article.

2. The concluding part of the abstract should be improved in terms of the results and future research directions to which this research can refer.

3. It is advisable to use more keywords to increase visibility and target more readers.

4. Avoid lumping the references. It is advisable to discuss them separately.

5.  The aim of the paper should be separated in the form of a last singular paragraph of the introduction and should be improved in terms of describing the contribution to the field under review and the elements of scientific novelty presented 

6. Figure titles should not be bolded. Review the instructions for authors in the journal template and modify accordingly.

7. Blanks between paragraphs are not required.

8. As the last paragraph of the Discussion section, it is advisable to detail the strengths, but more importantly the limitations of your study and to what extent these could be addressed for future research directions.

9.  It is also important to compare the results obtained with other similar studies.

10. How was the method of analysis validated? I refer to the validation criteria (accuracy, specificity, LOD, LOQ, etc.).  

11. More emphasis should be put on how to implement the results in clinical practice.

Author Response

REVIEWER 2

This article targets the role of the heteromeric CCR9:DRD5 complex in the migration of CD4+ T cells in the colonic mucosa upon inflammation, thus raising the therapeutic opportunity for treating IBD. The topic is relevant, but major deficiencies in form and content need to be addressed:

1. The similarity ratio shows a very high percentage (40%) of similarity with other works including the authors' own work. It is imperative that this percentage be significantly reduced, especially since it is an original research article.

ANSWER: The introduction and discussion have been extensively modified, so the similarity percentage should be significantly decreased.

2. The concluding part of the abstract should be improved in terms of the results and future research directions to which this research can refer.

 ANSWER: The concluding part of the abstract has been improved: “Our findings demonstrated that the CCR9:DRD5 heteromeric complex plays a fundamental role in the migration of CD4+ T-cells into the colonic mucosa upon inflammation. Thereby, the present study encourages the design of strategies for disassembling the formation of the CCR9:DRD5 as a therapeutic opportunity to treat IBD”.

3. It is advisable to use more keywords to increase visibility and target more readers.

 ANSWER: We have added three new keywords: G protein-coupled receptors, heteromers, inflammatory bowel diseases.

4. Avoid lumping the references. It is advisable to discuss them separately.

 ANSWER: Grouped references have been split out when possible. In some cases, it was not possible to split them out since they support the same information.

5.  The aim of the paper should be separated in the form of a last singular paragraph of the introduction and should be improved in terms of describing the contribution to the field under review and the elements of scientific novelty presented.

ANSWER: We have separated the aim of the study in a single paragraph (last paragraph in the introduction) and have added some lines describing the contribution of the results to the field: “In this study, we aimed to analyse the molecular requirements involved in the heteromer assembly as well as to determining the underlying cellular mechanisms involved in the colonic tropism given by the stimulation of the CCR9:DRD5 complex. The results obtained here improve our understanding on the complex mechanisms involving heteromeric receptors associated with leukocyte migration in inflammation”.

6. Figure titles should not be bolded. Review the instructions for authors in the journal template and modify accordingly.

 ANSWER: Figure titles are not bolded in the new version of the paper.

7. Blanks between paragraphs are not required.

ANSWER: Blanks between paragraphs have been removed.

8. As the last paragraph of the Discussion section, it is advisable to detail the strengths, but more importantly the limitations of your study and to what extent these could be addressed for future research directions.

ANSWER: We have included the limitations of our study and future perspectives for this research in the last paragraph of discussion.

9.  It is also important to compare the results obtained with other similar studies.

 ANSWER: We included a new paragraph (the fourth paragraph of the discussion) in which we discuss similar studies involving the role of GPCRs heteromers in the context of inflammatory and autoimmune disorders.

10. How was the method of analysis validated? I refer to the validation criteria (accuracy, specificity, LOD, LOQ, etc.).

ANSWER: We have included some lines about the validation criteria at the end of sections 4.3, 4.5, 4.7. 

11. More emphasis should be put on how to implement the results in clinical practice.

ANSWER: At the end of the discussion we included some lines about the future clinical directions for this research.

Reviewer 3 Report

Comments and Suggestions for Authors

Dear authors

The idea of the study is good.

Introduction is written well. But the references in the introduction need to update to year 2024. The newest reference is in year 2021 and many references are in 2002 to 2019.

In line 70 please delete recently because the study is in year 2021.

Results and discussion are logic and introduced well. The results are discussed very well.

The authors need to add the limitations of the study to enhance its value in the end of the discussion part. 

Materials and Methods part: Excellent.

Conclusion is too good.

Author Response

REVIEWER 3

Dear authors

The idea of the study is good. 

Introduction is written well. But the references in the introduction need to update to year 2024. The newest reference is in year 2021 and many references are in 2002 to 2019.

In line 70 please delete recently because the study is in year 2021.

Results and discussion are logic and introduced well. The results are discussed very well.

The authors need to add the limitations of the study to enhance its value in the end of the discussion part.

Materials and Methods part: Excellent. 

Conclusion is too good.

ANSWER: We thank the reviewer for his/her nice and constructive comments. We have updated the introduction with newer references when possible. However, in most cases, the references could not be replaced with newer ones, as they were seminal studies for the citation information. In the citation of our study from 2021 we have changed “We recently found…” with “We previously found…”. In addition, we have included the limitations of our study in the last paragraph of discussion.

Round 2

Reviewer 1 Report

Comments and Suggestions for Authors

Dear Authors,

Thank you for addressing my previous concerns thoroughly and revising the manuscript accordingly. I have two additional suggestions regarding the language of the newly added lines. In the sentence, “how it happens at the molecular level is not understood” (lines 60–64), the phrase “how it happens” seems too informal for a scientific manuscript. I suggest using more precise scientific language, such as “the molecular mechanisms underlying this process remain unclear.” Similarly, in the sentence “This is probably the reason of why vercirnon...” (lines 347–348), the phrasing “the reason of why” can be improved for clarity and scientific rigor. I recommend revising this to “This likely explains why vercirnon…” to maintain a more polished tone.

The rest of the manuscript looks great.

Comments on the Quality of English Language

The quality of the English language in the manuscript is generally good. However, there are a couple of areas where the phrasing could be improved to ensure clarity and scientific rigor.

Author Response

REVIEWER 1

Dear Authors,

Thank you for addressing my previous concerns thoroughly and revising the manuscript accordingly. I have two additional suggestions regarding the language of the newly added lines. In the sentence, “how it happens at the molecular level is not understood” (lines 60–64), the phrase “how it happens” seems too informal for a scientific manuscript. I suggest using more precise scientific language, such as “the molecular mechanisms underlying this process remain unclear.” Similarly, in the sentence “This is probably the reason of why vercirnon...” (lines 347–348), the phrasing “the reason of why” can be improved for clarity and scientific rigor. I recommend revising this to “This likely explains why vercirnon…” to maintain a more polished tone.

The rest of the manuscript looks great.

ANSWER: We thank the reviewer for his/her suggestions. We have modified both sentences as proposed by the reviewer.

Lines 61-63: “Although the signalling of DRD1 and DRD2 is dominant over DRD3 and DRD5 at high dopamine levels, the molecular mechanisms underlying this process remain unexplored”.

Lines 330-335: “This likely explains why vercirnon, a small molecule designed to inhibit CCR9 signalling, did not display significant beneficial effects for the treatment of CD (46). This probably also explains why the treatment of CD patients with natalizumab, a monoclonal antibody that blocks the interaction of a4 integrin with its ligand, was accompanied by collateral effects such as the risk of developing progressive multifocal leukoencephalopathy (46)”.

Reviewer 2 Report

Comments and Suggestions for Authors

The authors have significantly improved the manuscript based on the suggestions received.

Author Response

REVIEWER 2

The authors have significantly improved the manuscript based on the suggestions received.

ANSWER: We thank the reviewer for his/her comments that have contribute to improve the quality of our manuscript.